# Male Knock-in Mice Expressing an Arachidonic Acid Lipoxygenase 15B (Alox15B) with Humanized Reaction Specificity Are Prematurely Growth Arrested When Aging

**DOI:** 10.3390/biomedicines10061379

**Published:** 2022-06-10

**Authors:** Marjann Schäfer, Kumar R. Kakularam, Florian Reisch, Michael Rothe, Sabine Stehling, Dagmar Heydeck, Gerhard P. Püschel, Hartmut Kuhn

**Affiliations:** 1Department of Biochemistry, Charité—Universitätsmedizin Berlin, Corporate Member of Freie Universität Berlin and Humboldt Universität zu Berlin, Charitéplatz 1, 10117 Berlin, Germany; marjann.schaefer@googlemail.com (M.S.); kumar.kakularam@charite.de (K.R.K.); florian.reisch@charite.de (F.R.); sabine.stehling@charite.de (S.S.); dagmar.heydeck@charite.de (D.H.); 2Institute for Nutritional Sciences, University Potsdam, Arthur-Scheunert-Allee 114-116, 14558 Potsdam, Germany; 3Lipidomix GmbH, Robert-Rössle-Straße 10, 13125 Berlin, Germany; michael.rothe@lipidomix.de

**Keywords:** eicosanoids, lipid peroxidation, oxidative stress, polyenoic fatty acids, erythropoiesis

## Abstract

Mammalian arachidonic acid lipoxygenases (ALOXs) have been implicated in cell differentiation and in the pathogenesis of inflammation. The mouse genome involves seven functional *Alox* genes and the encoded enzymes share a high degree of amino acid conservation with their human orthologs. There are, however, functional differences between mouse and human ALOX orthologs. Human ALOX15B oxygenates arachidonic acid exclusively to its 15-hydroperoxy derivative (15*S*-HpETE), whereas 8*S*-HpETE is dominantly formed by mouse Alox15b. The structural basis for this functional difference has been explored and in vitro mutagenesis humanized the reaction specificity of the mouse enzyme. To explore whether this mutagenesis strategy may also humanize the reaction specificity of mouse Alox15b in vivo, we created *Alox15b* knock-in mice expressing the arachidonic acid 15-lipoxygenating Tyr603Asp+His604Val double mutant instead of the 8-lipoxygenating wildtype enzyme. These mice are fertile, display slightly modified plasma oxylipidomes and develop normally up to an age of 24 weeks. At later developmental stages, male *Alox15b*-KI mice gain significantly less body weight than outbred wildtype controls, but this effect was not observed for female individuals. To explore the possible reasons for the observed gender-specific growth arrest, we determined the basic hematological parameters and found that aged male *Alox15b*-KI mice exhibited significantly attenuated red blood cell parameters (erythrocyte counts, hematocrit, hemoglobin). Here again, these differences were not observed in female individuals. These data suggest that humanization of the reaction specificity of mouse Alox15b impairs the functionality of the hematopoietic system in males, which is paralleled by a premature growth arrest.

## 1. Introduction

Arachidonic acid lipoxygenases (ALOX isoforms) form a family of oxygen-metabolizing enzymes that convert polyunsaturated fatty acids into corresponding hydroperoxy derivatives [1,2,3,4]. The mouse genome involves seven functional *Alox* genes and most of them are localized in a joint *Alox* gene cluster on chromosome 11 [5]. In the human genome an orthologous gene exists for each mouse Alox isoform and most of these genes are clustered on the short arm of chromosome 17 [6]. When arachidonic acid (AA) is used as the substrate, mouse Alox isoforms exhibit different reaction specificities and AA 12-lipoxygenating (Alox15 [7] Alox12 [8], Alox12b [9], Aloxe12 [10], Aloxe3 [11]), AA 5-lipoxygenating (Alox5, [12]) and AA 8-lipoxygenating (Alox15b, [13]) enzymes have been reported. Despite the high degree of amino acid conservation (>80%), several human and mouse ALOX orthologs show remarkable functional differences. For instance, under physiological conditions, human ALOXE3 does not exhibit a fatty acid oxygenase activity [14], but under hyperoxic conditions the formation of dioxygenation products has been reported [15]. On the other hand, under normoxic conditions, mouse Aloxe3 was identified as AA 12-lipoxygenating enzyme [11]. Mouse Alox15 catalyzes AA 12-lipoxygenation [7], but the human ortholog produces 15-H(p)ETE as the dominant AA oxygenation product [16]. The structural basis for the distinct reaction specificities of mouse and human ALOX15 orthologs has been explored and critical amino acid residues have been identified [17,18,19]. Human ALOX15B converts AA almost exclusively to 15-HETE [20], but mouse Alox15b, which is called Alox8 according to the AA-based ALOX nomenclature, catalyzes AA 8-lipoxygenation [13,21]. Here again, the structural basis for the different reaction specificities of human and mouse ALOX15b orthologs has been explored and sequence determinants have been identified [22]. When Tyr603 and His604 of recombinant mouse Alox15b were mutated to the amino acids, which are present at these positions in human ALOX15B (Asp and Val, respectively), the corresponding double mutant catalyzed almost exclusively the formation of 15*S*-HETE [22]. When an inverse mutagenesis strategy was applied for human ALOX15B, the reaction specificity of this enzyme was shifted in favor of AA 8S-lipoxygenation [22]. These data suggested that Tyr603 and His604 might function as sequence determinants for the reaction specificity of these enzymes [22].

In addition to different reaction specificities, mouse and human ALOX15B orthologs show different tissue-specific expression profiles. Northern blot analyses indicated that human ALOX15B is mainly expressed in lung and prostate but no ALOX15B transcripts were detected in commercial mRNA preparations of spleen, thymus, testis, ovary, small intestine, colon, mixed leukocytes, heart, brain, placenta, liver, skeleton muscle, kidney and pancreas [20]. Mouse Alox15b mRNA was found at high concentrations in brain. At lower levels this mRNA was also present in heart, but it was not detected in spleen, lung, liver, skeletal muscle, kidney and testis [21]. In normal mouse skin, Alox15b mRNA was also detected in small amounts. However, after treatment with phorbol myristic acid (PMA) a strong stimulation of Alox15b expression was observed [13,21]. According to these data, PMA-treated mouse skin was the richest source of Alox15b.

To find out whether Tyr603Asp+His604Val exchange also modifies the reaction specificity of mouse Alox15b in vivo and to explore the functional consequences of this humanization of the reaction specificity of the enzyme, we created knock-in mice (*Alox15b*-KI mice), which express the AA 15-lipoxygenating Tyr603Asp + His604Val double mutant of mouse Alox15b instead of the AA 8-lipoxygenating wildtype enzyme. These mice are viable, breed normally and female individuals display similar body weight kinetics as outbred wildtype controls. In contrast, male *Alox15b*-KI mice gain significantly less body weight than wildtype controls when aging and this effect was paralleled by compromised erythropoiesis. Taken together, these data suggest that in vivo humanization of the reaction specificity of mouse Alox15b attenuated the functionality of the erythropoietic system in males, which is paralleled by a gender-specific premature growth arrest.

## 2. Materials and Methods

*Chemicals*—The chemicals used for this study were obtained from the following sources: Arachidonic acid (AA) and authentic HPLC standards of HETE-isomers (15*S*-HETE, 15*S/R*-HETE, 12*S/R*-HETE, 12*S*-HETE, 8R/S-HETE, 8S-HETE, 5*S/R*-HETE, 5*S*-HETE) from Cayman Chem. (distributed by Biomol GmbH, Hamburg, Germany); acetic acid from Carl Roth GmbH (Karlsruhe, Germany); sodium borohydride from Life Technologies, Inc. (Eggenstein, Germany); isopropyl-β-thiogalactopyranoside (IPTG) from Carl Roth GmbH (Karlsruhe, Germany); calcium ionophore A23187 and butylhydroxy toluene (BHT) from Merck (Darmstadt, Germany); restriction enzymes from ThermoFisher (Schwerte, Germany); the *E. coli* strain Rosetta2 DE3 pLysS from Novagen (Merck-Millipore, Darmstadt, Germany). Oligonucleotide synthesis was performed at BioTez Berlin Buch GmbH (Berlin, Germany). Nucleic acid sequencing was carried out at Eurofins MWG Operon (Ebersberg, Germany). HPLC grade methanol, acetonitrile, n-hexane, 2-propanol and water were from Fisher Scientific (Waltham, MA, USA). Phorbol 12-myristate 13-acetate (PMA) was purchased from PeproTech (Hamburg, Germany). The origins of other chemicals employed in this study are specified in the description of the methods for which they have been used. For oxylipidomic measurements, the following chemicals were used: Deuterated standards (LTB4-d4, 20-HETE-d6, 15-HETE-d8, 13-HODE-d4, 14,15-DHET-d11, 9,10-DiHOME-d4, 12,13-EpOME-d4, 8,9-EET-d11, PGE2-d4; 10 ng/mL each) from Cayman Chem. (Ann Arbor, MI, USA); acetonitrile, methanol and acetic acid from Merck (Darmstadt, Germany); ethyl acetate, n-hexane, NaOH, Na_2_HPO4, KH_2_PO4 from Fisher Scientific (Schwerte, Germany)

*Bacterial expression of mouse Alox15b variants*—Wildtype and mutant mouse Alox15b were expressed in *E. coli* as N-terminal his-tag fusion proteins. For this purpose, the coding region of the mouse *Alox15b* cDNA was cloned into the bacterial expression plasmid pET28b. Recombinant expression and preparation of the enzymes were performed as described before [23,24] and crude bacterial cell lysate supernatants were used as enzyme source.

*Site-directed mutagenesis of mouse Alox15b*—To humanize the reaction specificity of mouse Alox15b, site-directed mutagenesis was carried out using the PfuUltra II Hotstart PCR Master Mix kit (Agilent Technologies Germany GmbH & Co. KG, Waldbronn, Germany) as described before [23]. To create the Tyr603Asp+His604Val double mutant, the following mutation primers were synthesized. Upstream primer: 5′-GTT AAT TCG TCA AGT GAT GTC ATC ATT GCT CTC TGG-3′; downstream primer: 3′-CCA GAG AGC AAT GAT GAC ATC ACT TGA CGA ATT AAC-5′.

*In vitro activity assays of recombinant mouse Alox15b variants*—To assay the catalytic activities of the recombinant enzymes, variable amounts of cell lysate supernatants were added to 0.5 mL of PBS containing arachidonic acid (AA) as substrate at a final concentration of 100 µM. Incubation and analysis of the reaction products were performed with RP-HPLC as described before [23]. To resolve the hydroxy fatty acid enantiomers, combined normal phase/chiral phase HPLC (NP/CP-HPLC) was carried out. For this purpose, the conjugated dienes formed were purified with RP-HPLC and further analyzed by combined normal phase/chiral phase HPLC. For these analyses, a Chiralpak AD-H column (4.6 × 250 mm, 5 µm particle size, Daicel, Osaka, Japan) was connected with a Nucleosil pre-column (4.6 × 30 mm, 5 µm particle size, Macherey-Nagel, Düren, Germany) and the analytes were eluted isocratically using a solvent system consisting of n-hexane/methanol/ethanol/acetic acid (96/3/1/0.1, by vol.) at a flow rate of 1 mL/min.

*Creation of Alox15b-KI mice*—The *Alox15b*-KI mice characterized in this study were created in collaboration with Cyagen Bioscience (Santa Clara, CA, USA). To modify the *Alox15b* gene (GenBank accession number: NM_009661.4; Ensemble: ENSMUSG00000020891) that is located on mouse chromosome 11, we introduced two consecutive point mutations (Tyr603Asp + His604Val) into the wildtype mouse *Alox15b* gene using the Crispr/Cas9 mutagenesis strategy. The mouse *Alox15b* gene involves fourteen exons and its ATG start codon is located in exon 1, but the TAA stop codon is in exon 14. The target amino acids Tyr603 and His604 are both located in exon 13. For in vivo mutagenesis, the following gRNA targeting vectors and the donor oligonucleotide (with targeting sequence, flanked by 130 bp homologous sequences on both sides) were designed: (i) gRNA1 (matches forward strand of the gene): GTT ATC ACA TCA TTG CTC TCT GG (https://www.vectorbuilder.com/vector/VB171128-1087vxv.html, accessed on 10 May 2022); (ii) gRNA2 (matches reverse strand of the gene): AAC TTG ACG AAT TAA CTG CTG GG (https://www.vectorbuilder.com/vector/VB171128-1088vvs.html, accessed on 10 May 2022); (iii) donor oligonucleotide sequence: ACT ACT TCC AAA GGC CAG GCC CGG CCT GGA TTT CAT AGC CAG CTG CCA CAT TAA TTC GTC AAG TGA CGT CAT CAT TGC TCT CTG GCT GCT AAG CGC AGA ACC TGG GAC CAA GTA AGT AAG GAG CTG GGA. The mutation sites Tyr603Asp and His604Val (TAT CAC to GAC GTC) were introduced into exon 13 of the *Alox15b* gene by homology-directed repair. Binding and recutting of the sequence after homology-directed repair was prevented by introduction of a silent mutation (CTC to CTG) (Figure 1). gRNA generated by in vitro transcription and the donor oligonucleotide as well as the Cas9 mRNA were co-injected into fertilized eggs for the production of knock-in mice. The pups were genotyped with PCR followed by nucleotide sequence analysis of the amplification product. To reduce the possibility that our mutagenesis strategy introduced major off-target alterations into the mouse genome, we carried out off-target analyses. For this purpose, the mouse genome was screened in silico for the targeting sequence GTT ATC ACA TCA TTG CTC TCT GG and five potential off-target sites were identified. To exclude off-target alterations in these regions, genomic PCR was carried out and sequencing of the PCR products did not reveal any additional mutations (Appendix A).

*Genotyping*—The target region of the mouse *Alox15b* gene locus was amplified with PCR (annealing temperature 60 °C) with specific primers (mouse Alox15b forward: 5′-CGG GAA GCC CTG GTC CAG TAT ATC-3′; mouse Alox15b reverse: 5′-AGC CTC ACC CTG CCT CTA CTC TAA GT-3′) and the 656 bp PCR product was sequenced using the forward primer 5′-ATA TTC ACC TGC TCA GCC AAG CAT G-3′ to confirm in vivo mutagenesis.

*Ex vivo activity assays using PMA-treated mouse skin*—To explore whether our in vivo mutagenesis strategy altered the reaction specificity of mouse Alox15b, we carried out ex vivo activity assays. For this purpose, we used PMA-treated mouse skin as the enzyme source. To obtain the skin we sacrificed three wildtype mice and three *Alox15b*-KI animals, removed the tails and incubated them for 2 h in PBS containing 5 µM PMA. Afterwards, we prepared the epidermis and extracted total RNA from the proximal 3 cm. The remaining tissue was cut into small pieces and homogenized in 1 mL of PBS using a Fast-Prep-24 sample preparation system (MP Biomedicals, Irvine, USA). Cell debris was spun down and 0.6 mL of the homogenate supernatant was mixed with 0.4 mL of PBS containing 100 µM AA as Alox substrate. After a 30 min incubation period at room temperature, the reaction products were reduced by the addition of solid sodium borohydride, the sample was acidified by the addition of 35 µL acetic acid and the formed eicosanoids were extracted twice with 1 mL of ethylacetate. The extracts were combined, the solvent was evaporated under vacuum and the remaining lipids were reconstituted in 250 µL of acetonitrile. After vortexing for 2 min, 250 µL of water and 5 µL of acetic acid were added, the sample was centrifuged to remove insoluble material and 300 µL of the supernatant was injected for the RP-HPLC analysis. As for the in vitro activity assays (see above), a Shimadzu instrument (LC20 AD) equipped with a diode array detector (SPD M20A) was used and the hydroxy fatty acids were separated on a Nucleodur C18 Gravity column (Macherey-Nagel, Düren, Germany; 250 × 4 mm, 5 μm particle size) coupled with a guard column (8 × 4 mm, 5 μm particle size). A solvent system consisting of acetonitrile:water:acetic acid (70:30:0.1, by vol.) was employed at a flow rate of 1 mL/min and analytes were eluted isocratically at 25 °C. To resolve the hydroxy fatty acid enantiomers, combined normal phase/chiral phase HPLC was carried out. For this purpose, the conjugated dienes formed were purified with RP-HPLC and further analyzed by combined normal phase/chiral phase HPLC. For these analyses, a Chiralpak AD-H column (4.6 × 250 mm, 5 µm particle size, Daicel, Osaka, Japan) was connected with a Nucleosil pre-column (4.6 × 30 mm, 5 µm particle size, Macherey-Nagel, Düren, Germany) and the analytes were eluted isocratically using a solvent system consisting of n-hexane:methanol:ethanol:acetic acid (96:3:1:0.1, by vol.) at a flow rate of 1 mL/min.

*Determination of the basic hematological parameters*—Basic blood parameters (Hb, HK, erythrocyte count, leucocyte count, MCV, MHC, MCHC) of the two genotypes in three different age groups (young mice, 10–20 weeks; middle-aged mice, 30–40 weeks; old mice, 70–80 weeks) of either sex were determined. These analyses were performed at the Institut für Veterinärmedizinische Diagnostik GmbH (Berlin, Germany).

*Quantification of body weight kinetics*—Male and female *Alox15b*-KI mice and outbred wildtype control animals (*n* = 10 in each experimental group) were housed in separate cages (5 mice/cage) with water and standard chow diet ad libitum. The body weights were taken once a week over the time period indicated. The body weight kinetics were visualized using the GraphPad Prism version 8.2.0 for Windows (GraphPad Software, San Diego, CA, USA,) and statistic evaluation was carried out using the Wilcoxon test.

*Osmotic resistance of erythrocytes*—To assess the resistance of the red blood cells for osmotic stress, we followed the experimental protocol described before [25]. For this purpose, 2 μL of blood was diluted in 200 μL phosphate buffer (pH 7.4) containing NaCl at different concentrations (0–0.85%) and the cells were incubated for 30 min at room temperature. The samples were centrifuged at 200× *g* for 10 min and the supernatant containing the free hemoglobin, which was released during hemolysis, was quantified measuring the absorbance at 540 nm. The absorbance values were expressed as percentage of complete hemolysis, which was achieved when the cells were incubated in the absence of NaCl. This absorbance was defined as 100% hemolysis.

*Oxylipidomics*—To explore whether humanization of the reaction specificity of mouse Alox15b may impact the pattern of free plasma oxylipins, we quantified the amounts of free oxygenated PUFAs in the blood plasma. These data do not include the esterified derivatives. For this purpose, EDTA blood was drawn from sacrificed mice by heart puncture and the blood plasma was prepared by centrifugation. Then, 10 µL of blood plasma was mixed with 450 µL of water and 10 µL of a mixture of internal standards (LTB4-d4, 20-HETE-d6, 15-HETE-d8, 13-HODE-d4, 14,15-DHET-d11, 9,10-DiHOME-d4, 12,13-EpOME-d4, 8,9-EET-d11, PGE2-d4; 10 ng/mL each) and 5 µL butylhydroxytoluene (BHT) were added. Plasma proteins were precipitated by the addition of 100 µL of a 1:4 mixture (by vol.) of glycerol:water and 500 µL acetonitrile. The pH was adjusted to 6.0 by the addition of 2 mL phosphate buffer (0.15 M), the precipitated proteins were removed by centrifugation and the clear supernatant was used for solid phase lipid extraction on a 200 mg Agilent Bond-Elute-Certify II cartridge (Agilent Technologies, Santa Clara, CA, USA). Before sample application, the cartridge was conditioned with 3 mL methanol and 3 mL phosphate buffer (0.15 M, pH 6.0). After the sample was applied, the column was washed with 3 mL of a 1:1 mixture (by vol.) of methanol:water and the oxygenated fatty acids were eluted with a 74:25:1 mixture (by vol.) of ethyl acetate:n-hexane:acetic acid. The solvents were evaporated in a stream of nitrogen and the remaining lipids were reconstituted in 100 µL of a 6:4 mixture (by vol.) of methanol:water and used for LC-MS/MS analysis.

LC-MS/MS was carried out on an Agilent 1290/II LC-MS system consisting of a binary pump, an autosampler and a column oven (Agilent Technologies, Waldbronn, Germany). As stationary phase, we employed an Agilent Zorbax Eclipse C18 UPLC column (150 × 2.1 mm, 1.8 µm particle size). The temperature was set at 30 °C. As mobile phase, we used a solvent gradient that was mixed out of two stock solutions. Stock A: Water containing 0.05% acetic acid. Stock B: 1:1 mixture (by vol.) of methanol:acetonitrile. The solvent gradient employed for our analyses is specified in Appendix A. To avoid chromatographic artefacts, the injection system was rinsed after each injection with a 5:4:1 mixture (by vol.) of methanol:water:isopropanol. The HPLC system was connected with a triple quadrupole MS system (Agilent 6495 System, Agilent Technologies, Santa Clara, CA, USA). Negative electrospray ionization was carried out and the ionization parameters are given in Appendix A. The mass spectrometer was run in dynamic MRM-mode and each metabolite was detected simultaneously by two independent mass transitions (Appendix A). Experimental raw data were evaluated with the Agilent Mass-Hunter software package, version B10.0. For all metabolites analyzed in this study, individual calibration curves were set up (Appendix A) and the lower detection limits were determined (Appendix A).

*Ex vivo Alox5 activity assay*—EDTA blood was removed from sacrificed *Alox15b*-KI mice and outbred wildtype controls (*n* = 5 for each genotype). A 200 µL amount of the whole blood was incubated for 15 min at 37 °C in the absence or presence of 5 µM calcium ionophore A23187, which activates the Alox5 pathway. After the incubation period, the blood cells were removed by centrifugation and the plasma was quickly shock frozen in liquid nitrogen. Oxylipins were extracted and the LTB4 concentrations were quantified with LC-MS (see “Oxylipidomics” in the Material and Methods section).

*Statistics*—Statistic evaluation of the activity data and quantification of the patterns of AA oxygenation products was carried out with the two-sided Student’s *t*-test using the Microsoft Excel software package (Excel 2016) or the unpaired *t*-test using the GraphPad prism program. Numeric *p*-values <0.05 were considered statistically significant. Fertility data and blood parameters were analyzed using the Mann–Whitney U-test and body weight kinetics and osmotic resistance using the Wilcoxon test performed with the GraphPad Prism software package, version 8.2.0 for Windows (GraphPad Software, San Diego, CA, USA).

## 3. Results

*Tyr603Asp + His604Val exchange of mouse Alox15b humanized the reaction specificity of the recombinant enzyme*—To confirm that the reaction specificity of mouse Alox15b catalyzed AA oxygenation is humanized by Tyr603Asp + His604Val exchange, we first expressed wildtype mouse Alox15b and its Tyr603Asp + His604Val double mutant as recombinant N-terminal his-tag fusion proteins in *E. coli* and quantified the pattern of conjugated dienes formed during a 10 min incubation period of the recombinant enzymes with 100 µM arachidonic acid. As indicated in Figure 2A, conjugated dienes (inset to Figure 2A) co-migrating with authentic standards of 12-HETE and 8-HETE were formed by the recombinant wildtype enzyme. Smaller amounts of 15*S*-HETE were also detected. It should be stressed at this point that in our standard HPLC system 12-HETE and 8-HETE were not well resolved and thus, we analyzed the reaction product of the wildtype enzyme with an LC-MS method, in which the two critical HETE-isomers are well separated (inset to Figure 2B). Here, we identified the major AA oxygenation product as 8-HETE. Taken together, our HPLC and LC-MS analyses of the reaction products of recombinant wildtype mouse Alox15b indicated that 8-HETE is the major conjugated diene formed by this enzyme. Moreover, in the wildtype enzyme incubation, we observed small amounts of 5-HETE (Figure 2A). Similar amounts of 5-HETE were also detected in no-enzyme control incubation (data not shown) and thus this compound must be classified as an AA auto-oxidation product that was already present in the substrate stock solution.

In contrast, 15*S*-HETE was the major conjugated diene formed from AA by the Tyr603Asp + His604Val double mutant (Figure 2B). To obtain independent evidence for the chemical structure of the major AA oxygenation product formed by the mutant enzyme, the conjugated dienes formed were also analyzed with LC-MS. Here, we confirmed that 15-HETE was the major AA oxygenation product formed by the mutant enzyme (inset in Figure 2B). Statistic evaluation of the RP-HPLC raw data is given in Figure 2C. In summary, our analytical data confirmed [22] that wildtype mouse Alox15b oxygenates AA dominantly to 8-HETE, whereas the Tyr603Asp + His604Val exchange humanized the reaction specificity of this enzyme.

*Corresponding in vivo mutagenesis of the Alox15b gene humanized the reaction specificity of the enzyme*—To explore whether the reaction specificity of mouse Alox15b can also be humanized when an identical mutagenesis strategy was employed in vivo, we created knock-in mice expressing the Alox15b Tyr603Asp + His604Val double mutant instead of the wildtype enzyme using the Crispr/Cas9 strategy. For this purpose, a gRNA targeting vector as well as donor oligonucleotides were designed, which involved the targeting sequence flanked by 130 bp homologous sequences on both sides. The Tyr603Asp+His604Val (TAT-CAC to GAC-GTC) mutation sites were introduced into exon 13 by homology-directed repair mechanisms. In addition, a silent mutation (CTC to CTG) was introduced upstream of the targeting sequence to prevent the binding and re-cutting of the sequence after homology-directed repair (Figure 1). The Cas9 mRNA, the gRNA generated by in vitro transcription and the donor oligonucleotide were co-injected into fertilized eggs for the production of knock-in mice. The resulting pups were genotyped with PCR followed by sequence analysis of the amplification products. Heterozygous founder animals were mated, homozygous wildtype controls as well as homozygous knock-in mice were selected and colonies of *Alox15b* knock-in mice as well as outbred wildtype control animals were established.

Mouse Alox15b is expressed at high levels in PMA-treated skin [21]. To re-explore the tissue-specific expression pattern of mouse Alox15b, we first extracted total RNA from different tissues of wildtype mice (no PMA treatment) and quantified with qRT-PCR the expression levels of Alox15b mRNA. Using our qRT-PCR system we detected the highest expression levels in skin and lung but we also observed low-level expression of the enzyme in other tissues such as liver, kidney and bone marrow (Figure 3A). Since previous literature reports indicated that expression of Alox15b is strongly augmented in skin by treatment with PMA [13,21], we decided to use PMA-treated skin as the enzyme source for ex vivo activity assays. To prepare PMA-treated skin, we sacrificed *Alox15b*-KI mice and outbred wildtype controls, removed the tails and incubated them for 2 h at room temperature in PBS containing 5 µM PMA. After washing in PBS, the epidermis was prepared, total RNA was extracted from the proximal part (3 cm) of the tail epidermis and qRT-PCR was carried out to quantify the steady-state concentrations of the mRNAs encoding for the seven mouse Alox isoforms. From Figure 3B, it can be seen that in wildtype mice Alox12b, Alox15b and Aloxe3 are expressed at higher levels when compared with Alox12 and Alox12e. A similar expression pattern was observed in *Alox15b*-KI mice, but here elevated mRNA levels were observed for Alox12, Aloxe12 and Alox15b when wildtype animals were compared with the Alox15-KI mice. For Alox15 and Alox5, we did not detect significant mRNA concentrations in PMA-treated tail skin (data not shown). The most interesting result for the present study was that expression of Alox15b in *Alox15b*-KI mice was higher (about 3-fold) than in wildtype controls. Similar differences were also observed for Alox12 and Aloxe12, but the molecular bases for these findings have not been explored. For Alox15b, the increased expression levels in the *Alox15b*-KI mice might be interpreted as a compensatory response of the skin towards the lacking 8-HETE formation. Although a direct repressive effect of 8-HETE on the expression of the *Alox15b* gene has not been reported, it might be part of a feedback-control mechanism.

To explore whether our *Alox15b*-KI mice express an Alox15b with humanized reaction specificity, we next performed ex vivo activity assays. For this purpose, the epidermis of the PMA-treated mouse tails was homogenized in PBS (*n* = 3 for each genotype). After centrifugation (10,000× *g*), 0.6 mL of the homogenate supernatant was taken and 0.4 mL of PBS containing 100 µM of AA was added. After 30 min at room temperature, the reaction products were reduced, extracted and analyzed with RP-HPLC. From Figure 4A, it can be seen that conjugated dienes co-eluting with authentic standards of 12- and 8-HETE were formed by PMA-treated epidermis of wildtype mice. In addition, small amounts of 5-HETE were also detected, but combined NP/CP-HPLC indicated a racemic mixture, which was already present in the substrate solution. In contrast, formation of 15-HETE was hardly detected. Identical analyses of the reaction products formed by PMA-treated epidermis of *Alox15b*-KI mice also revealed the dominant formation of 12- and/or 8-HETE. Both compounds were not well resolved under these chromatographic conditions (Figure 4A). Importantly, in these activity assays we observed a clear 15-HETE peak (Figure 4B). When we compared the shape of two 8/12-HETE peaks formed by PMA-treated epidermis of wildtype and *Alox15*-KI mice, it became evident that the peak formed by wildtype epidermis was broader and not as sharp as that formed by *Alox15b*-KI epidermis. These data suggest that the wildtype peak might represent a mixture of two or more constituents, and eventually a mixture of 12- and 8-HETE. To explore the chemical structure of the reaction products in more detail, we prepared the conjugated dienes with RP-HPLC and further analyzed the products by combined normal-phase/chiral-phase HPLC (NP/CP-HPLC). From Figure 4C, it can be seen that the 8-HETE/12-HETE peak formed by wildtype epidermis consisted of a 3:1 mixture of 12*S*- and 8*S*-HETE, but we did not detect any 15*S*-HETE. In contrast, the major conjugated dienes formed by *Alox15b*-KI epidermis were identified as 12*S*-HETE and 15*S*-HETE (Figure 4D). Here, no 8*S*-HETE was detected. Quantification of the product patterns and statistical evaluation of the data are shown in Figure 4E. Taken together, the results of our ex vivo activity assays indicate that the major AA oxygenation product formed by PMA-treated epidermis was 12*S*-HETE. The metabolic origin of this product was not explored in this study. It might well be that Alox12, which is expressed in mouse skin, might be responsible for the formation of this product and experiments with Alox12 knockout mice would help to shed light on this open question. The 8*S*-HETE formation by wildtype epidermis is likely related to the expression of Alox15b. The lack of 15*S*-HETE formation by wildtype epidermis but the detection of this compound in the product mixture formed by *Alox15b*-KI epidermis is consistent with our hypothesis that our genetic manipulation of the *Alox15b* gene humanized the reaction specificity of this enzyme in vivo. This functional ex vivo activity data is consistent with the results of our genotyping strategy (insets in Figure 4C,D).

*Reproduction characteristics of Alox15b-KI mice*—*Alox15b*-KI mice are viable and reproduce normally. When we compared these animals with outbred wildtype controls, we did not observe significant differences between the two genotypes, when litter size (pups per litter), frequency of pregnancy (litters per female × month), number of pups per month (pups per female and month), number of premature casualties before weaning and the gender ratio of the newborns were compared (Figure 5). After birth, the newborns of both genotypes developed normally and there was no obvious evidence for post-partal developmental defects of the *Alox15b*-KI mice. In humans, ALOX15B is expressed at high levels in the skin and the enzyme was originally cloned from human hair roots [20]. As in humans, in mice, this enzyme is also expressed in skin [13,21] and thus humanization of its reaction specificity might have affected fur development and maintenance. However, when we evaluated fur development, we did not observe major differences between the two genotypes.

*Body weight kinetics of Alox15b-KI mice*—When we compared the kinetics of the absolute body weights of male and female *Alox15b*-KI mice with those of outbred wildtype controls starting 10 weeks after birth, we did not observe a significant difference (Wilcoxon test, *p* = 0.0607) between the two genotypes when female individuals were followed (Figure 6A). In fact, the growth curves were almost identical. Similar growth kinetics were also observed for male individuals during the first 24 weeks of post-partal development (Figure 6B) and statistic evaluation of the absolute body weights did not reveal significant differences between the two genotypes (Wilcoxon test, *p* = 0.6633) during this developmental period. However, at later time points the curves deviated from each other (Figure 6B) and highly significant differences (Wilcoxon test, *p* < 0.0001) were observed between the two genotypes. From these data, one may conclude that male *Alox15b*-KI gained significantly less body weight than outbred wildtype controls when aging and thus they experienced a premature growth arrest.

*Hematological parameters of Alox15b-KI mice*—In principle, there are multiple reasons for the observed premature growth arrest and a slightly compromised hematopoietic system might be one of them. To test the functionality of the hematopoietic systems of *Alox15b*-KI mice and outbred wildtype controls, we first compared the basic blood parameters (Hb, HK, erythrocyte count, leukocyte count, MCV, MHC, MCHC) of the two genotypes in three different age groups (young mice, 10–20 weeks; middle-aged mice, 30–40 weeks; old mice, 70–80 weeks) of either sex. Although all hematological parameters were in the normal range of the determination method used here, we found that for aged *Alox15b*-KI mice the major red blood cell parameters (erythrocyte count, hematocrit, hemoglobin) were significantly lower than the corresponding parameters of outbred wildtype controls (Figure 7). For young and middle-aged males, such differences were not observed. Interestingly, we did not find such differences for female individuals, irrespective of their age. Thus, as for the body weight kinetics, we observed gender-specific differences between *Alox15b*-KI mice and outbred wildtype controls but it remains to be explored whether the dysfunctionality of the erythropoietic system is responsible for the developmental retardation of male *Alox15b*-KI mice. It should be stressed at this point that for most other hematological parameters (reticulocyte count, MCV, MHC, MCHC) we did not observe significant differences between the two genotypes (Appendix A). Only for middle-aged female individuals the leukocyte counts of Alox15b-KI mice were significantly elevated (Appendix A), but the functional relevance of this difference has not been explored. In summary, when compared with outbred wildtype controls, aged male *Alox15b*-KI mice suffer from a mild normochromic normocytic anemia. Such a type of anemia frequently develops in patients with a primary defect in the erythropoietic system [26]. If this is the case for *Alox15b*-KI mice, the in vivo life span of peripheral red blood cells should be reduced, which frequently induces reticulocytosis and splenomegaly. To explore whether *Alox15b*-KI mice suffer from splenomegaly, we compared the spleen weights of middle-aged (30–40 weeks) *Alox15*-KI mice and outbred wildtype controls but did not detect significant differences between the two genotypes (Appendix A). Moreover, we did not observe reticulocytosis in aged male *Alox15b*-KI mice (Appendix A).

To obtain independent evidence for this hypothesis, we compared the sensitivity of red blood cells towards ex vivo osmotic challenge. When erythrocytes are incubated in hypo-osmotic solutions, they take up water, swell and finally hemolyse. If one determines the degree of hemolysis at different salt concentrations, the osmotic resistance of the cells can be quantified and an impaired osmotic resistance may be considered as an indicator for defective erythrocyte membrane functionality [27]. We quantified the osmotic resistance of wildtype and *Alox15b*-KI erythrocytes in the three different age categories (young, 10–20 weeks; middle-aged, 30–40 weeks; old, 70–80 weeks) of female and male individuals. From Figure 8A, it can be seen that erythrocytes of young female mice exhibit a higher osmotic resistance than cells prepared from middle-aged and aged individuals. At a given NaCl concentration, the extent of ex vivo hemolysis of erythrocytes prepared from young animals was significantly lower than that determined for cells prepared from middle-aged and aged individuals. Between the two latter age categories, we did not observe differences in the degree of hemolysis. Most importantly, for female mice, we did not detect significant differences in the osmotic resistance between erythrocytes of *Alox15b*-KI mice and wildtype controls in either of the three age categories. However, when similar experiments were carried out with male individuals, the situation was somewhat different (Figure 8B). Here again, we found that erythrocytes prepared from young individuals were less susceptible to osmotic challenge than cells of older mice and we did not observe significant differences between *Alox15*-KI mice and outbred wildtype controls for young and middle-aged male individuals. However, erythrocytes of aged *Alox15b*-KI mice were more resistant against osmotic challenge than the red cells of outbred wildtype control animals. On the first view, this finding is inconsistent with our conclusion that the erythropoietic system of *Alox15b*-KI mice is compromised since, following this hypothesis, a reduced osmotic resistance would be expected for *Alox15b*-KI erythrocytes. However, endogenous compensation reactions may have occurred and we refer to this point in more detail in the “Discussion” section.

*Oxylipidomics*—To explore whether our genetic manipulation of the *Alox15b* gene impacted the patterns of the plasma oxylipins, we quantified the plasma levels of more than 40 different oxylipins (Appendix A). Twelve of them, including a number of maresins, resolvins and neuroprotectins, were below the detection limits of our analytical systems (Appendix A), but for 32 oxygenated fatty acid derivatives, we obtained reliable analytic data (Appendix A). For these experiments, blood plasma of five middle-aged male individuals of each genotype were explored and the most relevant findings are briefly discussed below.

When we summed up the different free oxylipins detected in the blood plasma of both *Alox15b*-KI mice and outbred wildtype controls, we found significantly more oxygenated PUFA derivatives in Alox15-KI mice (Figure 9A). This difference was mainly due to the significantly higher plasma concentrations of 12-HETE (Appendix A), 12-HETrE (Appendix A) and 13-HODE (Appendix A) in *Alox15b*-KI mice (compared with wildtype controls). These three compounds are the dominant free oxylipins in the plasma of both genotypes. Their metabolic origin remains a matter of discussion. Although our data suggest a role of Alox15b in their biosynthesis (humanization of the reaction specificity of Alox15b elevated the relative abundance of these metabolites), the mechanistic details have not yet been defined.

Mouse Alox15b converts AA mainly to 8S-HETE, but the dominant AA oxygenation product of the humanized enzyme was 15-HETE (Figure 2). If our genetic manipulation of the *Alox15b* gene is mirrored on the level of the oxygenated plasma lipids, one would expect to see elevated plasma concentrations of 15-HETE but reduced plasma levels of 8-HETE (*Alox15*-KI mice vs. wildtype controls). Unfortunately, such differences were not observed (Figure 9B). Similarly, we did not detect the expected changes in the plasma levels of the EPA oxygenation products (Figure 9C) since 8-HEPE was not reduced in *Alox15b*-KI mice and 15-HEPE was not augmented. Wildtype mouse Alox15b converts DHA mainly to 10-HDHA, but the humanized enzyme forms 17-HDHA as a major DHA oxygenation product (Appendix A). If these changes were mirrored on the level of the oxygenated plasma lipids, one would expect to see lower 10-HDHA levels but higher 17-HDHA levels in *Alox15b*-KI mice than in wildtype controls. However, we did not observe the expected differences (Figure 9D). When we finally analyzed the major oxygenation products of 8,11,14-eicosatetraenoic acid, we measured significantly higher 15-HeTrE levels in *Alox15b*-KI mice (Figure 9E) and this finding was predicted as a functional consequence of our genetic manipulation. On the other hand, we did not observe a concomitant decrease in the 8-HeTrE plasma levels (Figure 9E). Here again, we measured higher 8-HeTrE concentrations. Since for linoleic acid and alpha-linolenic acid, the oxygenation products formed by recombinant wildtype mouse Alox15b and its Tyr603Asp+His604Val double mutant have not been determined, it was impossible to predict the alterations in the plasma oxylipidomes (Appendix A).

Among the di- and tri-hydroxylated PUFA derivatives, we only detected 5S,12S-DiHETE (Appendix A) and 10R,17S-DiHDHA (NPx, Appendix A) as plasma oxylipins. However, there were no significant differences in the plasma concentration of these metabolites between the two genotypes. Other more complex oxylipins, such as 8S,15S-diHETE, RvE1, RvD1, RvD1(17R), RvD2, RvD3, RvD4(17epi), RvD5, Mar-1, Mar-2, Mar(7epi) and NPD1, were below the detection limits of our analytical method and thus we cannot comment on whether mouse Alox15b or its Tyr603Asp+His604Val double mutant might be involved in the biosynthesis of these metabolites.

Taken together, our lipidomic data suggest that humanization of the reaction specificity of mouse Alox15b induces subtle alterations in the blood plasma concentrations of several oxylipins. However, it remains to be explored how exactly the observed lipidomic changes may be related to our subtle genetic manipulation.

*Humanization of Alox15b specificity does not affect the Alox5 pathway*—Since Alox5 and Alox15b are co-expressed in myeloid cells, it might be possible that the two enzymes might directly or indirectly interact with each other. To explore whether humanization of the reaction specificity of Alox15b may impact the Alox5 pathway in peripheral blood cells, we carried out ex vivo Alox5 activity assays. For this purpose, we stimulated whole blood with calcium ionophore A23187 and quantified the formation of leukotriene B (LTB4) with LC-MS. From Figure 10 it can be seen that wildtype blood cells form large amounts of LTB4 when stimulated with A23187. In contrast, no LTB4 was formed in the absence of this stimulus. When similar incubations were carried out with whole blood prepared from *Alox15b*-KI mice, similar amounts of LTB4 were detected. Statistically, there was no significant difference between the two genotypes. Taken together, these data indicate that humanization of the reaction specificity of mouse Alox15b hardly impacts the Alox5 pathway of peripheral blood cells.

## 4. Discussion

*In vivo mutagenesis humanizes the reaction specificity of mouse Alox15b*—Mouse Alox15b oxidizes arachidonic acid mainly into 8*S*-H(p)ETE [13,21]. In contrast, the human ortholog constitutes an AA 15-lipoxygenating enzyme [20]. The structural basis for this functional difference has been explored in detail and in vitro mutagenesis studies on recombinant mouse Alox15b (Tyr603Asp + His604Val double mutant) humanized the reaction specificity of this enzyme [22]. To explore whether a similar mutagenesis strategy will humanize the reaction specificity in vivo, we created knock-in mice carrying a minimally mutated version of the *Alox15b* gene, which encodes for the Tyr603Asp + His604Val double mutant. For this purpose, we employed the Crispr/Cas9 technology and established a colony of homozygous *Alox15b*-KI animals expressing the Tyr603Asp + His604Val Alox15b double mutant instead of the wildtype enzyme as well as a colony of outbred wildtype control animals. To explore whether this minor genomic manipulation (exchange of five nucleotides) indeed induced humanization of the reaction specificity, we carried out ex vivo activity assays using phorbol ester treated mouse skin as an enzyme source. For these experiments, we incubated homogenates of PMA-treated tail epidermis of three individuals of each genotype with exogenous AA and analyzed the oxygenated AA derivatives by different types of HPLC. When wildtype epidermis was taken through this experimental protocol, 12*S*-HETE was detected as the dominant AA oxygenation product (Figure 4A). The metabolic source of this compound has not been explored, but arachidonic acid 12-lipoxygenating enzymes, such as Alox15, Alox12 and Aloxe12, might contribute. In addition, smaller amounts of 8S-HETE were detected (Figure 4C) and this metabolite most probably originated from PMA-induced epidermal Alox15b. When we used PMA-treated mouse skin of *Alox15b*-KI mice as the enzyme source, the major arachidonic acid oxygenation product was also 12*S*-HETE. In addition, we observed significant amounts of 15*S*-HETE (Figure 4B,D), which was not present when wildtype skin was used. 8*S*-HETE was hardly formed in these incubations (Figure 4A,C). Taken together, these data confirm the predicted functional consequences of our genetic manipulation and the results suggested that our in vivo mutagenesis strategy humanized the reaction specificity of mouse Alox15b.

*Alox15b-KI mice may help to explore the biological function of Alox15b*—The biological role of ALOX15B in humans and mice has not completely been clarified. Since the human enzyme is expressed at high levels in hair follicles, it has been implicated in hair growth [20]. When human peripheral monocytes were differentiated in vitro to macrophages, expression of ALOX15B was strongly augmented [28]. In fact, Western blot analysis indicated that interleukin-4 (IL4), bacterial lipopolysaccharide and hypoxia increased the expression of ALOX15B but not of ALOX12. Since IL4 drives the differentiation of naive monocytes to alternatively activated macrophages (M2 macrophages) and since M2 macrophages have been implicated in inflammatory resolution [29,30], ALOX15B may play a role in this process. A similar expression regulation has previously been reported for human ALOX15 [31,32], but the regulatory mechanisms appear to be different for the two human ALOX isoforms. ALOX15B and ALOX15 [33,34] have been implicated in the biosynthesis of specialized pro-resolving lipid mediators (SPMs) and this catalytic activity may be related to the pro-resolving activities of M2-macrophages. However, because of the different reaction specificities of the two ALOX15 orthologs [13,20,21], the biosynthetic mechanisms and the SPM profiles in mice and humans should be different. ALOX15B has also been implicated in the pathogenesis of hyperproliferative diseases [35]. In fact, in prostate cancer, expression of the *ALOX15B* gene is silenced and mechanistic studies suggested a function of the *ALOX15B* gene as a tumor suppressor gene [36,37,38]. During development of ovarian cancer, expression of ALOX15B is strongly augmented and high levels of ALOX15B mRNA have also been detected in metastatic tissue [39]. Although the mechanistic details for this expression regulation have not been explored, quantification of ALOX15 mRNA in the blood has been suggested as a marker for ovarian cancer [39]. Since mouse Alox15b exhibits a different reaction specificity than its human ortholog [13,20,21], it remains unclear whether mouse Alox15b fulfils similar functions in mouse models of prostate and/or ovarian cancer. For such studies, the *Alox15b*-KI mice expressing an enzyme with humanized reaction specificity might be useful as a mechanistic tool.

Male *Alox15b* knockout mice show impaired recovery from influenza infection when compared with wildtype littermates [40]. In fact, six-month-old *Alox15b^-/-^* mice displayed a significantly prolonged state of illness, as indicated by altered body temperatures, impaired locomotor activities and delayed body weight recovery [40]. Moreover, the steady-state concentrations of the pro-inflammatory cytokine interleukin 6 in the lungs were significantly elevated during the phase of inflammatory resolution [40]. Interestingly, such differences were not observed when 3-month-old individuals were taken through the experimental protocol. These data suggest that *Alox15b* deficiency compromised the immune response towards influenza virus infection in aged *Alox15b*-deficient mice. Obviously, this immunological defect was well compensated in 3-month-old individuals. Unfortunately, no female mice were used for these experiments so that a possible gender specificity could not be explored. In our study, we observed similar age-dependent alterations. The curves of body weight kinetics of male *Alox15b*-KI mice were superimposable with those of outbred wildtype controls (Figure 6B) up to an age of 24 weeks. Afterwards, the curves of the body weight kinetics deviated from each other and *Alox15b*-KI mice gained significantly less body weight than outbred wildtype controls. Interestingly, this difference was gender-specific for male individuals since the curves of the body weight kinetics of females remained superimposable up to an age of 64 weeks. The molecular basis for this gender-specific premature growth arrest has not been explored in detail but it might be related to our observation that the erythropoietic system of male Alox15-KI individuals is compromised in aged male *Alox15b*-KI mice (Figure 7).

*Humanization of the reaction specificity of mouse Alox15b induces malfunction of the erythropoietic system in aged male Alox15b-KI mice*—Although *Alox15b^-/-^* mice develop normally, their immune response towards influenza virus infection is compromised [40]. During the initial phase of this response, the innate immune system is activated at the site of infection and natural killer (NK) cells enter this site to eradicate virus-infected cells [41]. In addition, neutrophils and monocytes are recruited to the site of infection, helping to clear the tissue from infected cells [42]. The observation that *Alox15b^-/-^* mice need more time to recover from an experimental infection with the influenza virus [40] suggests that Alox15b may play a role for the normal functionality of the immune system. Unfortunately, the precise function of the enzyme has not been explored and hematopoietic parameters (leukocyte count, differential blood count) have not been reported. We found that the leukocyte counts of *Alox15b*-KI mice and of outbred wildtype controls of either sex were not significantly different (Appendix A). However, aged male *Alox15b*-KI mice expressing a mutant Alox15b isoform with humanized reaction specificity carry a compromised erythropoietic system (Figure 7). Interestingly, this erythropoietic defect was not observed in female individuals. For the time being, the molecular basis for the defective erythropoiesis in aged male *Alox15b*-KI mice remains unclear but work is in progress to shed light on this interesting aspect.

When we quantified the osmotic stability of wildtype erythrocytes and compared it with that of red blood cells prepared from *Alox15b*-KI mice, we observed an improved osmotic resistance of *Alox15b*-KI erythrocytes (Figure 8). On first view, this observation is inconsistent with our conclusion that the erythropoietic system of *Alox15b*-KI mice is compromised. Following this hypothesis, a reduced osmotic resistance would be expected for *Alox15b*-KI erythrocytes. However, it might be possible that the improved osmotic resistance of *Alox15b*-KI erythrocytes may be the result of an adaptive response of the erythropoietic system to balance the reduced capacity of the bone marrow to produce red blood cells. Osmotic-resistant erythrocytes are likely to live longer and this may compensate the attenuated erythropoietic capacity. It would be of interest to explore experimentally whether *Alox15b*-KI erythrocytes really show a prolonged in vivo life span.

*Gender-specific effects of humanization of the reaction specificity of mouse Alox15b*—As indicated in Figure 6 and Figure 7, humanization of the reaction specificity of mouse Alox15b induced a premature gender-specific growth retardation in male *Alox15b*-KI mice (Figure 6) and a gender-specific dysfunction of the erythropoietic system (Figure 7 and Figure 8) in aged individuals. Although the molecular mechanisms for the gender-specific character of these effects remain to be explored, our observation that female individuals do not show these effects suggests that both the premature growth retardation and the dysfunctionality of the erythropoietic system may not be the consequence of differences in the genetic background of the *Alox15b*-KI mice and wildtype animals. If background problems that can never be completely excluded were responsible for the observed defects, similar phenotypic alterations would also be expected for female individuals.

In humans, leukotriene-related diseases such as bronchial asthma and allergic rhinitis show an obvious gender bias. In fact, adult female individuals are more frequently affected than males [43]. Ex vivo formation of leukotrienes by female blood was significantly higher than that by male blood and the addition of 5-dehydrotestosteron to female blood reduced the ex vivo leukotriene biosynthetic capacity of female blood to male levels. These data indicated that androgens may regulate the catalytic activity of Alox5 and the underlying mechanisms have been reported [44]. Moreover, female individuals are more susceptible than males for anti-leukotriene therapy [45] and taken together these data indicate gender-specific differences in ALOX5 functionality. However, for the time being, gender-specific effects have not been reported for other human ALOX paralogs such as ALOX15, ALOX15B and ALOX12. Although the molecular basis for the gender-specific premature growth arrest of male *Alox15b*-KI mice has not been clarified, this is the first report suggesting that gender-specific effects may also be related to Alox15b.

## Figures and Tables

**Figure 1 biomedicines-10-01379-f001:**
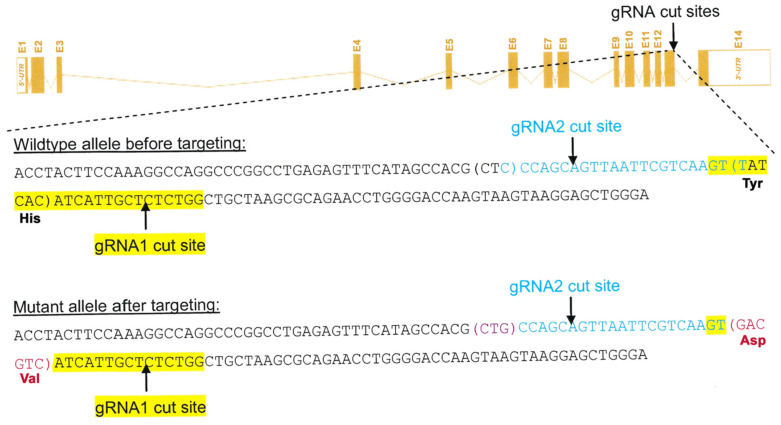
**Crispr/Cas9 strategy for creating Alox15b-KI mice.** To create *Alox15b*-KI mice, we employed the Crispr/Cas9 strategy to introduce the Tyr603Asp + His604Val double mutant into the wildtype *Alox15b* gene. The resulting knock-in mice express the AA 15-lipoxygenating Tyr603Asp + His604Val double mutant instead of the AA 8-lipoxygenating wildtype enzyme. For this purpose, the donor oligonucleotide (TAT-CAC to GAC-GTC) was introduced into exon 13 by homology-directed repair mechanisms. In addition, a silent mutation (CTC to CTG) was introduced.

**Figure 2 biomedicines-10-01379-f002:**
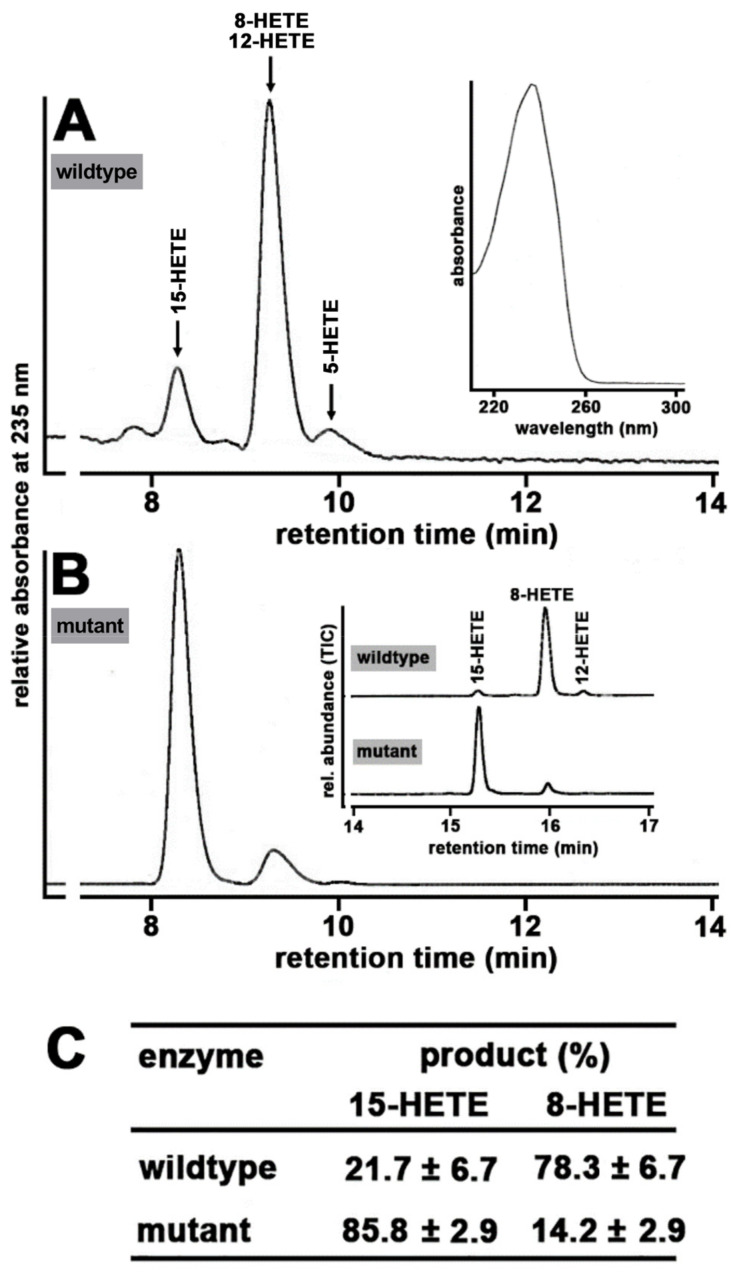
**Reaction specificity of wildtype and mutant recombinant mouse Alox15b.** Wildtype and mutant (Tyr603Asp + His604Val) Alox15b were expressed as N-terminal his-tag fusion proteins as described in Section 2 and aliquots of the bacterial lysis supernatants were used as enzyme source. After a 10 min incubation period with exogenous AA (100 µM), the reaction products were analyzed with RP-HPLC, as reported in Section 2. (**A**) Representative partial RP-HPLC chromatogram of the AA oxygenation products formed by wildtype mouse Alox15b. Retention time of authentic standards are indicated by the arrows above the traces. Inset: UV-spectrum of the conjugated dienes eluting with a retention time of 8.5 min. (**B**) Representative partial RP-HPLC chromatogram of the AA oxygenation products formed by the Tyr603Asp + His604Val double mutant of mouse Alox15b. Inset: LC-MS analyses of the AA oxygenation products formed by wildtype and mutant Alox15b. (**C**) Statistical evaluation of the major AA oxygenation products formed by wildtype and mutant mouse Alox15b (*n* = 4).

**Figure 3 biomedicines-10-01379-f003:**
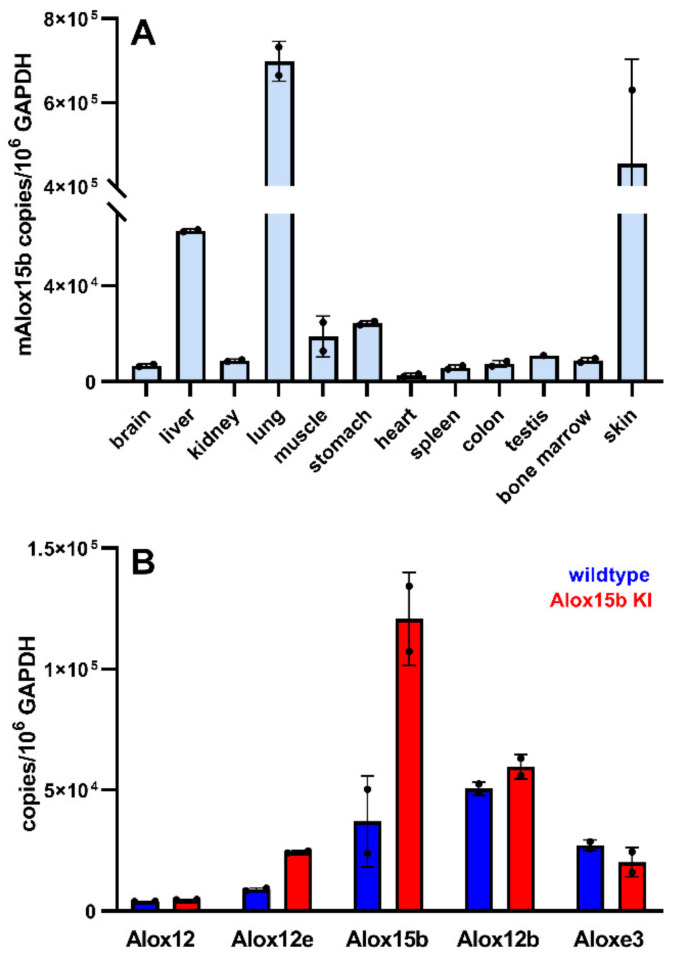
**Expression of Alox isoforms in different mouse tissues.** (**A**) Expression of Alox15b in different mouse tissues. Total RNA was extracted from different mouse tissues and Alox15b mRNA was quantified with qRT-PCR as described in Section 2. Two independent measurements (*n* = 2) were carried out for each RNA extract. (**B**) Expression of Alox isoforms in PMA-treated mouse skin. Total RNA was extracted from PMA-treated tail epidermis of *Alox15b*-KI mice (red columns) and outbred wildtype controls (blue columns). Expression of the mRNA of different Alox isoforms was quantified by qRT-PCR. Two independent measurements (*n* = 2) were carried out for each RNA extract. We also attempted to quantify expression of Alox15 and Alox5 but did not get specific PCR signals. Thus, these two Alox isoforms are not expressed on PMA-treated epidermis.

**Figure 4 biomedicines-10-01379-f004:**
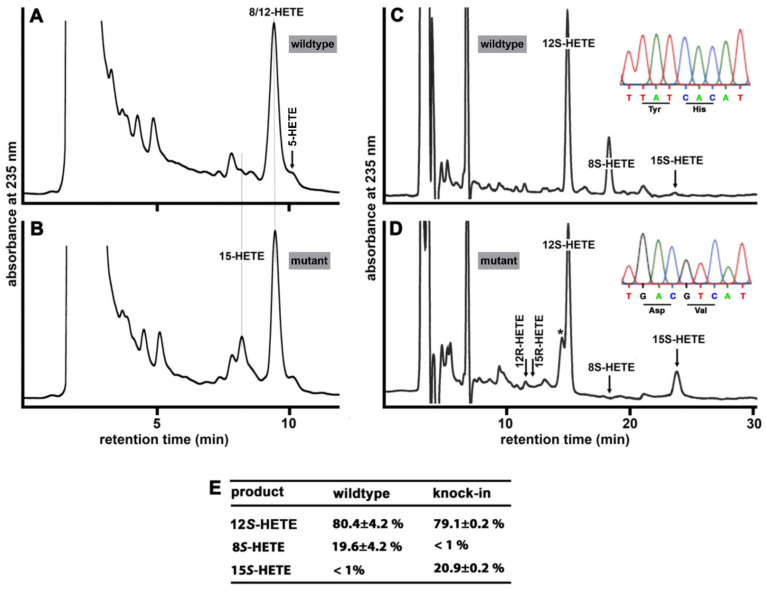
**Ex vivo Alox activity assays using PMA-treated tail epidermis as source of mouse Alox15b.** Expression of Alox15b was induced in mouse tail epidermis by treatment with PMA and ex vivo activity assays were carried out using aliquots of a homogenate supernatant of mouse tail epidermis (see Section 2). (**A**) RP-HPLC analysis of the product pattern formed from AA when PMA-treated tail epidermis of wildtype mice was used as enzyme source. (**B**) RP-HPLC analysis of the product pattern formed from AA when PMA-treated tail epidermis of *Alox15b*-KI mice was used as enzyme source. (**C**) The conjugated dienes formed by PMA-treated wildtype epidermis were prepared with RP-HPLC (panel A) and further analyzed by combined normal phase/chiral phase HPLC (see Section 2). Inset: Representative genotyping chromatogram. (**D**) The conjugated dienes formed by PMA-treated *Alox15b* KI epidermis were prepared with RP-HPLC (panel B) and further analyzed by combined normal phase/chiral phase HPLC (see Section 2). * This front shoulder peak did not show a conjugated diene chromophore. Inset: representative genotyping chromatograms. (**E**) Statistical evaluation of the major AA oxygenation products. For this evaluation, the experimental raw data of NP/CP-HPLC were used.

**Figure 5 biomedicines-10-01379-f005:**
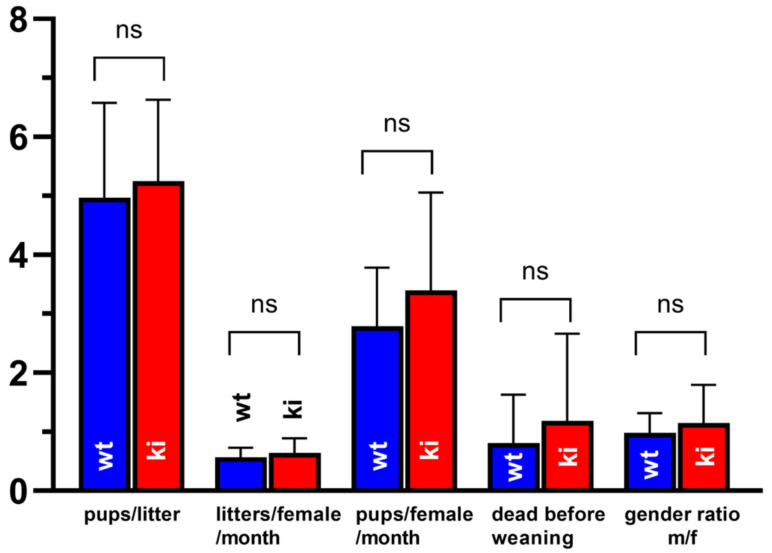
**Comparison of fertility parameters of *Alox15b*-KI mice and outbred wildtype controls.** For *Alox15b*-KI mice, 14 breeding pairs and for outcrossed wildtype controls, 18 breeding pairs (one male + 2 females) were mated and the different fertility parameters (*x*-axis) were quantified over a breeding period of 56 (*Alox15b*-KI mice) and 65 (wildtype controls) breeding months, respectively. ns, no significant difference.

**Figure 6 biomedicines-10-01379-f006:**
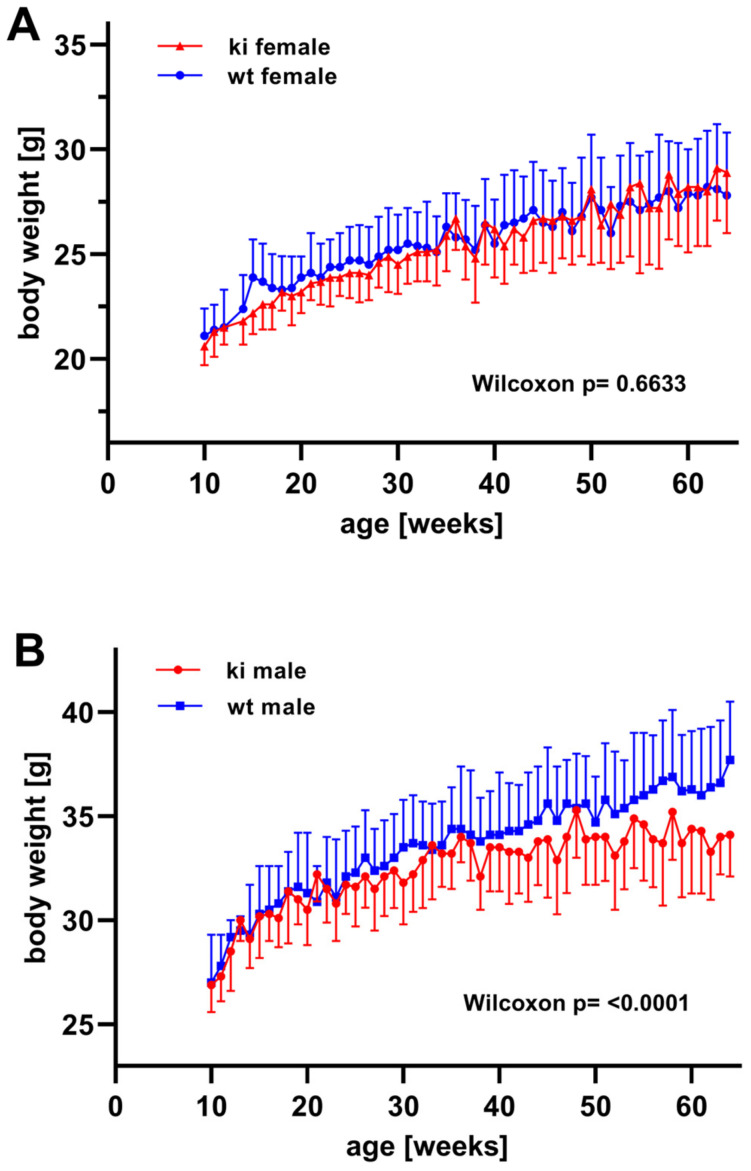
**Comparison of the body weight kinetics of *Alox15b*-KI mice and outbred wildtype controls.** For each genotype, 10 individuals of either sex were housed in separate cages (5 mice/cage) with water and food (standard chow diet) ad libitum. Absolute body weights were quantified once a week, covering the developmental period between 10 and 64 weeks. (**A**) Female individuals, (**B**) male individuals.

**Figure 7 biomedicines-10-01379-f007:**
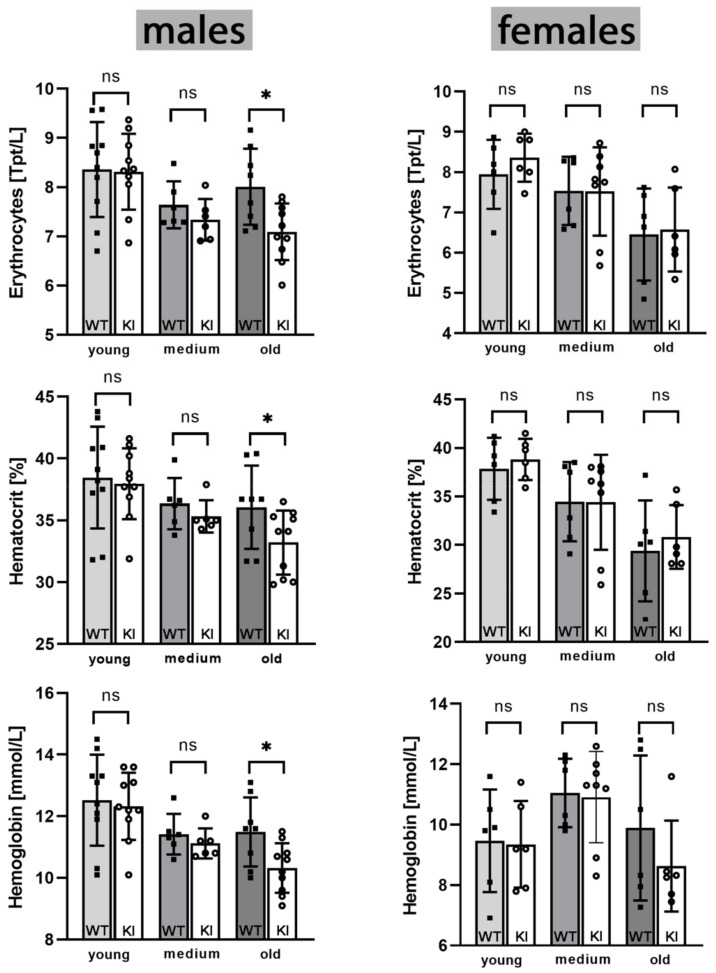
**Comparison of basic erythrocyte parameters of *Alox15b*-KI mice and outbred wildtype controls of either sex.***Alox15b*-KI mice and outbred wildtype controls of either sex were classified in three age categories (young mice, 10–20 weeks; middle-aged mice, 30–40 weeks; old mice 70–80 weeks, *n* ≥ 5 for each age group). After sacrificing the animals by cervical dislocation under anesthesia, EDTA blood was removed by heart puncture. The basic hematological parameters were determined by the Institut für Veterinärmedizinische Diagnostik GmbH (Berlin, Germany). Relevant erythrocyte parameters are shown in this image. The complete sets of hematological parameters are given in Appendix A. * indicates significant differences (*p* < 0.05). ns, no significant difference.

**Figure 8 biomedicines-10-01379-f008:**
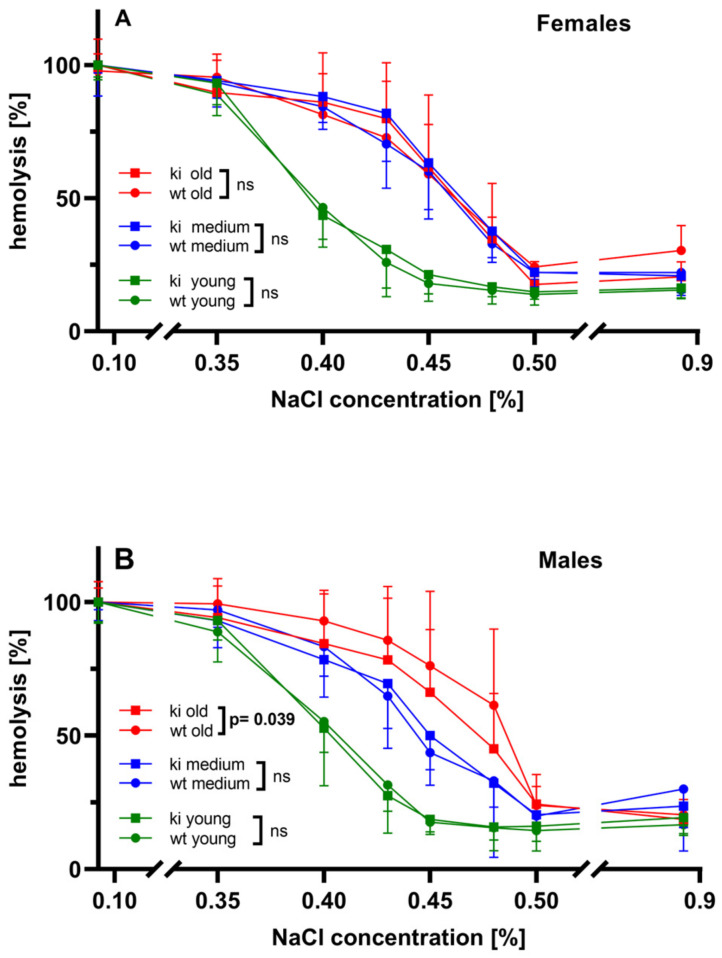
Comparison of the susceptibility of erythrocytes prepared from *Alox15b*-KI mice and from outbred wildtype controls for osmotic challenge. *Alox15b*-KI mice and outbred wildtype controls of either sex were classified in three age categories (young mice, 10–20 weeks; middle-aged mice, 30–40 weeks; old mice, 70–80 weeks, *n* ≥ 5 for each age-group with 2–3 replicates/individual). After sacrificing the animals by cervical dislocation under anesthesia, EDTA blood was removed and the susceptibility of the red blood cells for osmotic challenge was quantified as described in Section 2. The degree of hemolysis was calculated and these data are plotted over the NaCl concentration. Under strongly hypo-osmotic conditions (water, no NaCl), all erythrocytes (100%) hemolyzed. Under iso-osmotic conditions (0.85% NaCl), more than 80% of the red blood cells survived the hemolysis period. (**A**) Female mice, (**B**) male mice. ns, no significant difference.

**Figure 9 biomedicines-10-01379-f009:**
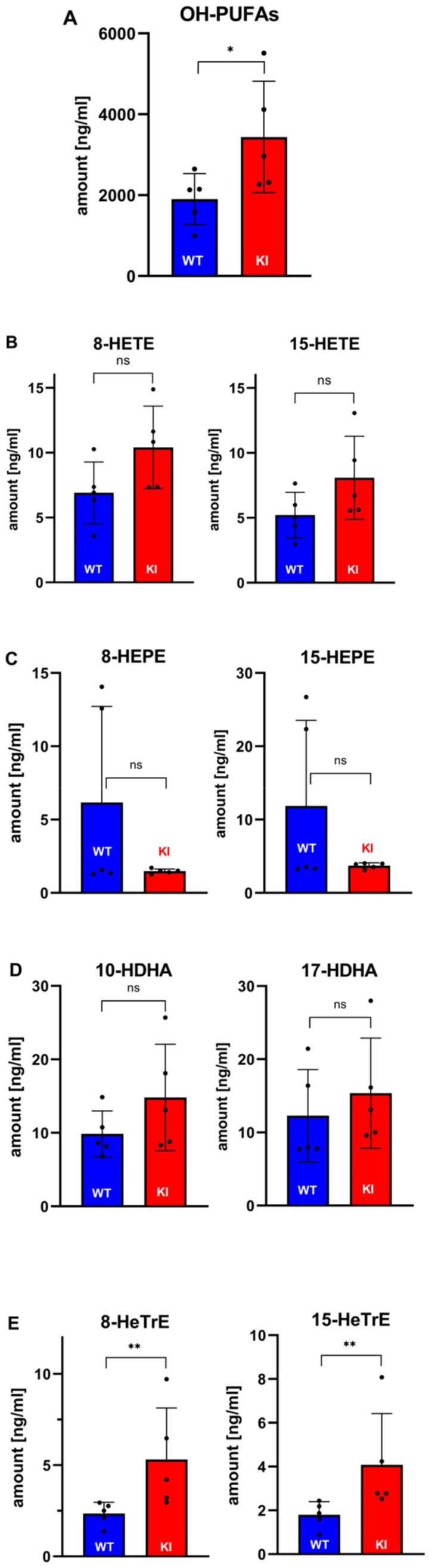
**Quantification of selected oxylipins in the blood plasma of *Alox15b*-KI mice and outbred wildtype controls.** Male *Alox15b*-KI mice and outbred wildtype controls (*n* = 5 for each genotype) were sacrificed, EDTA blood was removed, plasma lipids were extracted and the free plasma oxylipidomes were quantified with LC-MS (for methodological details see Section 2). Quantifications of selected metabolites are given. (**A**) Sum of all quantified oxylipins (OH-PUFAs). (**B**) Quantification of the most relevant arachidonic acid (AA) metabolites, (**C**) Quantification of the most relevant 5,8,11,14,17-eicosapentaenoic acid (EPA) metabolites. (**D**) Quantification of the most relevant 4,7,10,13,16,19-docosahexaenoic acid (DHA) metabolites. (**E**) Quantification of the most relevant 8,11,14-eicostrienoic acid metabolites. * indicates significant difference at *p* < 0.05, ** indicates significant differences at *p* < 0.01. ns, no significant difference.

**Figure 10 biomedicines-10-01379-f010:**
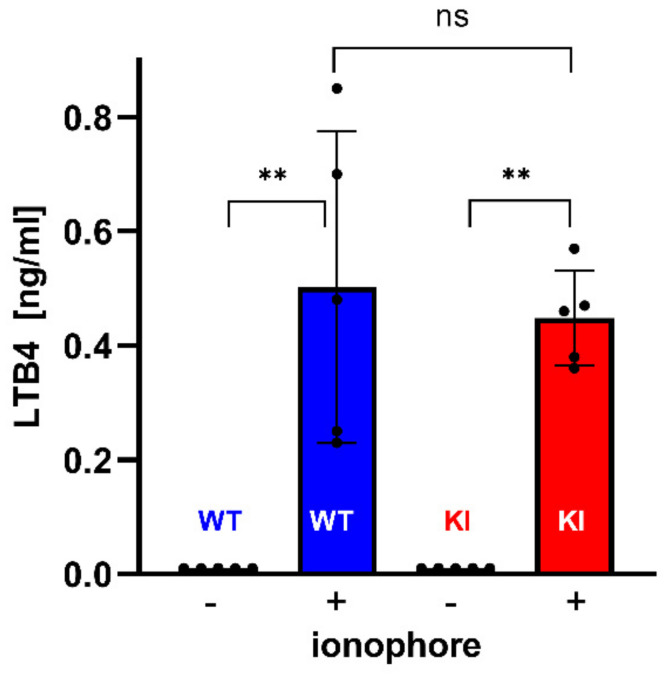
**Ex vivo Alox5 activity assays using whole blood of *Alox15b*-KI mice and outbred wildtype controls as experimental system**. Male *Alox15b*-KI mice and outbred wildtype controls (*n* = 5 for each genotype) were sacrificed, heparinized blood was collected and the Alox5 pathway was stimulated by incubating the blood with 5 µM of calcium ionophore A23178. Cells were spun down and the blood plasma was shock frozen in liquid nitrogen. For analysis, the oxylipins were extracted and the free plasma LTB_4_ levels were quantified with LC-MS (for methodological details see Section 2). ** indicate significant differences (*p* < 0.01). ns, no significant difference.

## Data Availability

The original experimental raw data can be obtained from the authors upon request.

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
