# Peer review of "Male Knock-in Mice Expressing an Arachidonic Acid Lipoxygenase 15B (Alox15B) with Humanized Reaction Specificity Are Prematurely Growth Arrested When Aging"

_biomedicines, 2022, doi:10.3390/biomedicines10061379_

Round 1
Reviewer 1 Report
I found this to be a very interesting examination of the impact of humanized reaction specificity on mice. I think this is a valuable contribution to the field. However, there are several issues that I believe need to be addressed:
Line 44, should Aox15 be Alox15 instead?
Line 80, should be Tyr603Asp+His604Val
In the HPLC and LC-MS analysis shown in figure 2, why was a 12-HETE standard only used in the LC-MS analysis? 5-HETE is shown in the HPLC, while only 12-HETE and not 5-HETE is shown in the LC-MS.
In Figure 2 the graph readability could be improved if a simple wildtype/mutant label was added to parts A and B (similar to the labels in the insert LC-MS graph)
Line 377, in all previous cases, the figure label was printed using bold font
In the examination of Alox expression (figure 3), data points were used to show the individual samples in figure 3A, but not in 3B. it would be helpful to include data points in 3B to show the variability between samples more clearly.
Adding wildtype/Alox15b-KI labels in Figure 4 would help make the figure more clear
In figure 6, only positive error bars are shown for blue, while negative error bars are shown for red. Is this for better visibility? If so, this should be clearly indicated in the caption.
I don’t like the axis break in figure 6, as it makes it difficult to see immediately what age the data starts at. The graph would only be extended slightly be removing this axis break.
I believe the bar graphs with data points, like in Figure 7 are much more valuable than standard bar graphs, as they show the full variability and range of the data samples. The simple bar graphs in Figures 5, 9, and 10 (and also many figures in the supplemental information) could be improved by adding data points similar to figure 7
In Figure 7 the caption states that there were 5 samples for each age-group. However, more than 5 data points are shown. Can you explain why this is the case?
Figure 8 would be more clear with a male and female label on the graph itself
Line 566, should read Female
Line 662, should this read 12S-HETE?
Line 668, the heading does not seem to match the content of the paragraph. I believe this heading should be moved down to the next paragraph.
The authors indicate that Alox15b-KO mice have compromised immune response to influenza virus. It would be interesting to see if the Alox15b-KI has a similar effect. Have the authors considered investigating this? This could be another potential impact of this humanized reaction specificity.
Line 727 should read leukocyte
Line 728 should read significantly
In figure 10, some of the differences appear quite large in the bar graph, but are not statistically significant. Is it possible that this is due to the low sample numbers (only n=5)? Again, if the data points were shown on the bar graphs it would be easier to see if this is simply due to a few outliers.
Line 762, should read genetic
Author Response
Dear Editor,
on behalf of all co-authors, I should like to resubmit the revised version of our MS entitled “Male knock-in mice expressing an arachidonic acid lipoxygenase 15b (Alox15b) with humanized reaction specificity are prematurely growth arrested when aging” for publication in Biomedicines. All co-authors would like to express our gratitude to the reviewer and the editor for valuable advice, which contributed to improve the quality of the paper. The critical points raised in the evaluation report were addressed on the point-to-point basis as indicated below:
Comment of reviewer/editor: I found this to be a very interesting examination of the impact of humanized reaction specificity on mice. I think this is a valuable contribution to the field. However, there are several issues that I believe need to be addressed:
Reply of authors: We thank the reviewer/editor for overall positive evaluation of our MS and for their helpful advice leading to amendment of the paper.
Comment of reviewer/editor: Line 44, should Aox15 be Alox15 instead?
Reply of authors: We apologize for this typo, that was corrected during revision.
Comment of reviewer/editor: Line 80, should be Tyr603Asp+His604Val
Reply of authors: We apologize for this mistake but we corrected it during revision.
Comment of reviewer/editor: In the HPLC and LC-MS analysis shown in figure 2, why was a 12-HETE standard only used in the LC-MS analysis? 5-HETE is shown in the HPLC, while only 12-HETE and not 5-HETE is shown in the LC-MS.
Reply of authors: In our standard HPLC system12-HETE and 8-HETE are not well separated. To clearly identify whether 12-HETE or 8-HETE is formed by the wildtype enzyme we additionally analyzed the reaction products by LC-MS and with this method the two critical isomers are well separated. In other words, our standard HPLC indicated that either 12-HETE or 8-HETE is the major product of wildtype Alox15b. Additional LC-MS analyses excluded major 12-HETE formation and thus indicated the major AA oxygenation product of the wildtype enzyme as 8-HETE. This is now clearly explained in the text (page 6-7). We also modified Figure 2 in such a way that we indicated in panel A that standards of 12-HETE and 8-HETE basically co-eluted under our experimental conditions. Moreover, in the wildtype enzyme incubation, we observed small amounts of 5-HETE. However, these small amounts of 5-HETE were also present in the no-enzyme control incubation and thus, the small amounts of 5-HETE must be classified as auto-oxidation product of AA. We also explained this in the revised ms.
Comment of reviewer/editor: In Figure 2 the graph readability could be improved if a simple wildtype/mutant label was added to parts A and B (similar to the labels in the insert LC-MS graph)
Reply of authors: We followed the advice of the reviewer/editor and added corresponding labels (wildtype, mutant) to Figure 2A+B.
Comment of reviewer/editor: Line 377, in all previous cases, the figure label was printed using bold font.
Reply of authors: For the revised MS we corrected this mistake.
Comment of reviewer/editor: In the examination of Alox expression (figure 3), data points were used to show the individual samples in figure 3A, but not in 3B. it would be helpful to include data points in 3B to show the variability between samples more clearly.
Reply of authors: We followed the advice of the reviewer/editor and restructured Figure 3. It now involves all data points in both panels.
Comment of reviewer/editor: Adding wildtype/Alox15b-KI labels in Figure 4 would help make the figure more clear
Reply of authors: We followed the advice of the reviewer/editor and added the requested labels to Figure 4.
Comment of reviewer/editor: In figure 6, only positive error bars are shown for blue, while negative error bars are shown for red. Is this for better visibility? If so, this should be clearly indicated in the caption.
Reply of authors: We tested both variants (positive + negative error bars vs. alternative error bars) and decided to use alternative error bars for better visibility. Using positive + negative error bars the error bars would overlay which impairs visibility.
Comment of reviewer/editor: I don’t like the axis break in figure 6, as it makes it difficult to see immediately what age the data starts at. The graph would only be extended slightly be removing this axis break.
Reply of authors: We modified Figure 6 as suggested by the reviewer/editor.
Comment of reviewer/editor: I believe the bar graphs with data points, like in Figure 7 are much more valuable than standard bar graphs, as they show the full variability and range of the data samples. The simple bar graphs in Figures 5, 9, and 10 (and also many figures in the supplemental information) could be improved by adding data points similar to figure 7.
Reply of authors: In Figure 5, 14 litters of Alox15b-KI mice and 18 litters of wildtype mice were analyzed. According to our opinion this number of data points is too big to be shown from as individual data points. This would make the image more difficult to read and thus, we would like to keep the image as it is. In Figures 9 and 10 the individual data points are now shown. For the Supplement we would like to keep the original figure versions since restructuring the images is rather laborious using the original raw data tables.
Comment of reviewer/editor: In Figure 7 the caption states that there were 5 samples for each age-group. However, more than 5 data points are shown. Can you explain why this is the case?
Reply of authors: This was a misprint. We corrected it to ≥ 5.
Comment of reviewer/editor: Figure 8 would be more clear with a male and female label on the graph itself.
Reply of authors: We modified the image accordingly.
Comment of reviewer/editor: Line 566, should read Female
Reply of authors: We corrected this error in the revised ms.
Comment of reviewer/editor: Line 662, should this read 12S-HETE?
Reply of authors: We corrected this error in the revised ms.
Comment of reviewer/editor: Line 668, the heading does not seem to match the content of the paragraph. I believe this heading should be moved down to the next paragraph.
Reply of authors: The reviewer is correct that heading and text do not match. To overcome this problem, we rephrased the heading for this paragraph to “Alox15b-KI mice may help to explore the biological function of Alox15b”.
Comment of reviewer/editor: The authors indicate that Alox15b-KO mice have compromised immune response to influenza virus. It would be interesting to see if the Alox15b-KI has a similar effect. Have the authors considered investigating this? This could be another potential impact of this humanized reaction specificity.
Reply of authors: This is an interesting point that is worth to be explored. However, such experiments require collaborative studies between two research groups. We neither have the methodological knowhow nor official permission to set up the influence infection model. We are of course willing to give our mice to any lab around the world to perform such experiments.
Comment of reviewer/editor: Line 727 should read leukocyte
Reply of authors: We corrected this error in the revised ms.
Comment of reviewer/editor: Line 728 should read significantly
Reply of authors: We corrected this error in the revised ms.
Comment of reviewer/editor: In figure 10, some of the differences appear quite large in the bar graph, but are not statistically significant. Is it possible that this is due to the low sample numbers (only n=5)? Again, if the data points were shown on the bar graphs it would be easier to see if this is simply due to a few outliers.
Reply of authors: In response of the reviewers comment we modified the outline of this image in such a way that all individual data points are now visible. Indeed, the big interindividual differences within the different experimental groups as well as the limited n-numbers (n=5) might contribute to the insignificant differences.
Comment of reviewer/editor: Line 762, should read genetic
Reply of authors: We corrected this error in the revised ms.
We included the following Institutional Review Board Statement: into the ms indicating that our animal experiments were carried out according to the guidelines of the declaration of Helsinki. “The study was conducted according to the guidelines of the Declaration of Helsinki and approved by the Institutional Review Board of the State Animal Care Committee (Landesamt für Gesundheit und Soziales, Berlin, Germany) and the following permission number was given: T0437/08.”
We hope that the revised version of the MS may now be suitable for publication in Biomedicines.
Sincerely
H. Kühn, D. Heydeck

Reviewer 2 Report
Mammalian arachidonic acid lipoxygenases (ALOX) are involved in cell differentiation and in the pathogenesis of inflammation. To elucidate the specificity of the murine Alox15b response in vivo, we created Alox15b knockout mice expressing the Tyr603Asp+His604Val double mutant, 15-lipoxygenating arachidonic acid, instead of the wild-type 8-lipoxygenating enzyme. The authors found that at later stages of development, male Alox15b-KI mice gained significantly less body weight than outbred wild-type controls, but this effect was not observed in females, while old male Alox15b-KI mice showed significantly reduced red blood cell parameters (number erythrocytes, hematocrit, hemoglobin). These data suggest that humanization of the specificity of the murine Alox15b response disrupts the functionality of the hematopoietic system in males, which is accompanied by premature growth arrest. The article is well written, the experiment is well planned, the research methods are described in detail, the illustrative material is harmoniously selected. I think that the article can be accepted for publication in its present form.
Author Response
Thank you for this positive evaluation of our ms. During the revision process we included some minor modifications as suggested by reviewer 1. The basic take-home message of the paper remained unaltered.
Sincerely
H. Kühn, D.Heydeck